# An Analysis of the Implementation and Use of (Critical) Incident Reporting Systems ((C)IRSs) in German Hospitals: A Retrospective Cross-Sectional Study from 2017 to 2022

**DOI:** 10.3390/healthcare12232386

**Published:** 2024-11-27

**Authors:** Carlos Ramon Hölzing, Patrick Meybohm, Charlotte Meynhardt, Oliver Happel

**Affiliations:** University Hospital Würzburg, Department of Anaesthesiology, Intensive Care, Emergency and Pain Medicine, Oberdürrbacher Street 6, 97080 Würzburg, Germany; meybohm_p@ukw.de (P.M.); meynhardt_c@ukw.de (C.M.); happel_o@ukw.de (O.H.)

**Keywords:** patient safety, healthcare quality improvement, human factors, human error, safety culture, incident reporting system, healthcare safety, retrospective study, standardisation, centralisation

## Abstract

**Background:** Incident reporting systems (IRSs) have become a central instrument for improving patient safety in hospitals. In Germany, hospitals are legally required to implement internal IRSs, while participation in cross-institutional IRSs is voluntary. **Methods**: In a retrospective, descriptive cross-sectional study, the structured quality reports of all German hospitals from 2017 to 2022 (2598–2408 hospitals (2017–2022)) were analysed. The participation of hospitals in internal and cross-institutional IRSs was examined, as was the frequency of training and evaluations of incident reports. **Results:** The rate of participation in internal IRSs increased from 94.0% in 2017 to 96.6% in 2019 and remained stable at 96.0% in 2022. About 85% of hospitals conducted internal evaluations of the incident reports, with monthly evaluations being the most common (33.9%). Training on how to use IRSs was mostly provided on an ad hoc basis (41.6% in 2022), with regular training being less common. Participation in cross-institutional IRSs increased significantly from 44.5% in 2017 to 55% in 2019 and remained stable until 2022. Participation in hospital IRSs showed significant increases, while specialised systems exhibited lower participation rates. **Conclusions**: Internal IRSs have been established in German hospitals; however, there is still room for improvement in conducting regular training sessions and evaluations. Although participation in cross-institutional IRSs has increased, it remains fragmented. Further centralisation and standardisation could enhance efficiency and contribute to an improvement in patient safety.

## 1. Introduction

The German healthcare system is characterised by a decentralised structure with a strong emphasis on solidarity and universality. Hospitals in Germany are categorised into three ownership types: public (owned by municipalities or federal states), private for-profit, and non-profit institutions, such as those run by charitable organisations. Funding is derived from a combination of insurance contributions, government subsidies, and out-of-pocket payments [1]. This diverse ownership and financing framework creates variations in hospitals’ priorities, resource allocation, and quality management, including the implementation of incident reporting systems (IRSs).

The development and implementation of incident reporting systems (IRSs) or critical incident reporting systems (CIRSs) in the German healthcare system constitutes a substantial stride towards enhancing patient safety and the quality of care [2,3]. These systems enable the systematic recording and analysis of critical incidents, thereby facilitating the development of preventive interventions [4]. While the term IRSs generally refers to systems designed to document and analyse adverse events, the abbreviation CIRSs (critical incident reporting systems), as used in this paper, denotes a specific subset focusing on incidents with significant harm potential. This distinction underlines the need for precise definitions and clarity in discussing their respective contributions to patient safety. Incident reporting systems (IRSs) play a critical role in improving patient safety and the quality of healthcare, offering a range of benefits that extend beyond individual institutions. They enable the systematic identification of system vulnerabilities, support organisational learning, and foster a culture of safety by encouraging transparency and accountability among healthcare professionals. Furthermore, data from IRS analyses can inform policy decisions and the development of best practices on both the institutional and national levels, helping to standardise safety protocols and reduce the recurrence of critical incidents.

The World Health Organisation (WHO) has identified the reporting of incidents and the use of IRSs as a pivotal strategy in the advancement of patient safety and quality of care. This is particularly emphasised in the WHO Global Action Plan on Patient Safety 2021–2030, which considers incident reporting to be an essential component in promoting global patient safety initiatives [5]. While studies in Germany emphasise incident reporting systems as a key strategy for patient safety, other efforts in the Gulf Cooperation Council Countries focus on patient-centred care and addressing disparities in healthcare access [6].

Globally, incident reporting systems have been implemented in various healthcare systems with differing levels of success. For example, countries such as the United Kingdom and the United States have long-established national reporting systems, like the National Reporting and Learning System (NRLS) and the Patient Safety Reporting System (PSRS), respectively. In contrast, in many low- and middle-income countries, the adoption of such systems faces significant barriers, including limited resources, a lack of infrastructure, and insufficient training for healthcare professionals [7]. Understanding these global variations underscores the necessity of tailoring incident reporting systems to specific healthcare contexts while learning from international best practices.

Despite their potential, IRSs face several challenges that hinder their effectiveness and acceptance. One of the primary obstacles is fostering an organisational culture that encourages employees to report incidents, as a significant proportion of professionals are reluctant to report critical incidents due to concerns about potential negative consequences for their careers or the reputation of their institution [8,9]. In addition, inadequate training of staff in the effective use of these systems can result in employees either not using them or even rejecting them [10]. Finally, the support of the organisation itself is essential for the success of an IRS [10]. Furthermore, underreporting remains a pervasive issue, often driven by fear of repercussions, a lack of awareness, or the perception that reporting does not lead to meaningful change [7,9]. Additionally, the absence of standardised data formats and insufficient interoperability between systems can hinder the timely exchange and comprehensive analysis of critical incidents, further limiting their impact on patient safety [11].

In accordance with § 135a SGB V, hospitals are legally required to engage in quality assurance measures, which necessitates the implementation of risk management protocols. These protocols mandate the establishment of internal IRSs [12]. Although there is no explicit federal law requiring participation in cross-institutional IRSs, it is strongly encouraged and forms part of the structured quality reports submitted by hospitals. In 2016, the German legislature introduced financial incentives to promote participation in cross-institutional IRSs, aiming to increase hospital engagement [11]. Hospitals retain the flexibility to choose from one of the 13 cross-institutional IRSs that are currently recognised by the Federal Joint Committee (G-BA) for case submission. Insufficient interoperability between different systems can result in important information being either unrecorded or unduly delayed in being exchanged, which in turn impairs the overall effectiveness of IRSs [13].

Despite these challenges, there is a lack of comprehensive analyses that explore the prevalence of, utilisation of, and barriers to IRSs in Germany. This study aims to address these gaps by providing empirical insights that can inform strategies for standardisation and improved engagement. The sensitivity of this work lies in its potential to directly impact patient safety by identifying systemic weaknesses and facilitating preventive measures. It also highlights the importance of creating an environment where healthcare professionals feel safe and supported in reporting incidents. Addressing these issues is not only critical for improving patient outcomes but also for fostering a culture of trust and accountability within healthcare organisations.

The objective of this study is to provide a descriptive analysis of the utilisation and prevalence of both internal and cross-institutional IRSs in Germany while contextualizing these findings within the broader global landscape. By focusing on trends, barriers, and facilitators, our research sheds light on the current state of IRS participation and highlights opportunities for standardisation and enhanced engagement. Through this analysis, we aim to contribute to the understanding of how IRSs can be optimised to support patient safety and quality care in Germany and beyond.

## 2. Materials and Methods

The present study is a retrospective, descriptive cross-sectional study. The aim of the investigation was to analyse the implementation and use of institutional and cross-institutional IRSs in German hospitals. The manuscript was prepared in accordance with the STROBE checklist [14].

### 2.1. Setting and Participants

This study was based on the structured quality reports of all hospitals in Germany that were approved according to § 108 SGB V, which legally requires hospitals to submit annual quality reports. All hospitals and specialised clinics that are mandated to provide these reports were included in the analysis, ensuring comprehensive coverage of the healthcare landscape. The analysis used data from reports spanning the period from 2017 to 2022, with the exception of 2018. Unfortunately, the reports for 2018 were not made available by the Federal Joint Committee (G-BA) without further explanation. The data were analysed between August and October 2024. The study included all hospitals that were licensed according to § 108 SGB V, which meant that all hospitals that were legally obliged to submit a structured quality report were included in the analysis.

Geographic locations were classified into regions based on postal codes (e.g., East Germany, Northeast Germany, Middle Germany, etc.), and descriptive statistics for bed count were calculated for each year. In 2017, the number of hospital beds ranged from 0 to 5692, with a mean of 268.35 (SD = 315.14). In 2022, the average number of beds was 214.04 (SD = 250.311). The geographic distribution of hospitals across regions and over time is summarised in Table 1.

### 2.2. Variables

The study focused on two aspects: internal and cross-institutional incident reporting systems.

Internal IRSs: The variables examined were the frequency of meetings, documentation and procedural instructions, internal evaluations of the reports received, and training of employees in how to use the incident reporting system. An internal incident reporting system was considered to be present if at least one of the following variables was documented: documentation, internal evaluation, or training.

Cross-institutional incident reporting systems: Participation in the cross-institutional incident reporting systems approved by the G-BA was analysed. These include the CIRS AINS (German Society of Anaesthesiology and Intensive Care (DGAI), German Medical Association), CIRS Berlin (Berlin Medical Association, German Medical Association), Hospital CIRS (German Hospital Federation, German Nursing Council, German Medical Association), CIRS NRW (North Rhine and Westphalia Medical Associations, North Rhine-Westphalia Hospital Association, North Rhine and Westphalia-Lippe, Federal Medical Association), CIRS Emergency Medicine (Department of Anaesthesia, Intensive Care and Emergency Medicine, Kempten Clinic), CIRS of the German Society of Surgery, CIRS of the German Pain Society (DGS), ‘Every mistake Counts (Institut für Allgemeinmedizin Frankfurt), DokuPIK (Documentation of pharmaceutical interventions in hospitals of the Federal Association of German Hospital Pharmacists (ADKA)), CIRS Health Care, PaSIS (Patient Safety and Information System of the University Hospital Tübingen), DGHO-CIRS (German Society of Haematology and Medical Oncology), and others. A hospital was considered to be participating in a cross-institutional incident reporting system if it was involved in at least one of these systems.

### 2.3. Statistical Methods

The statistical analysis of the data was conducted using the IBM SPSS Statistics software, version 29. The graphical representation was created using Prism 9^®^ (GraphPad Software Inc., Boston, MA, USA). The nominal variables were described using absolute frequencies and percentages. Comparative analyses for nominally scaled variables were performed using the Chi-square test (X^2^) to evaluate differences between groups. When an expected cell frequency was less than five, Fisher’s exact test was applied to ensure robust and accurate statistical inference. The null hypothesis posits equality between the groups, whereas the alternative hypothesis suggests a difference between the groups. A *p*-value of less than 0.05 was considered statistically significant for all analyses.

## 3. Results

The participation rate of hospitals in the internal incident reporting system was 94.0% (n = 2442) in 2017, 96.6% (n = 2514) in 2019, and 96.0% (n = 2312) in 2022 (Table 2 and Figure 1).

The implementation of internal evaluations of incident reports demonstrated a relatively constant trend over the years (Table 3). In 2017, 82.7% of hospitals conducted evaluations, rising to 86.2% in 2019 and remaining stable around 85% through 2022. The proportion of hospitals not conducting evaluations decreased from 17.3% in 2017 to 14.5% in 2022.

Incident reports have been evaluated on a monthly basis in approximately 33% of hospitals (2017: 32.2%; 2022: 33.9%). Quarterly evaluations were conducted by approximately 17% of hospitals (2017: 16.6%; 2022: 16.6%). In 2022, 23% of evaluations were conducted on a need-based basis, representing a slight increase from the previous year (2017: 22.8%). A smaller proportion of hospitals conducted half-yearly and annual evaluations (2022: 4.4% half-yearly, 7.5% yearly). No reports were received concerning the performance of evaluations on a weekly basis.

Training of employees in the use of the incident reporting system was observed in 70.9% of hospitals (n = 1842) in 2017, which remained largely unchanged in the following years (Table 4). Training on the incident report system was most often provided on an as-needed basis, with values between 37.5% in 2017 and 41.6% in 2022. Subsequently, annual training was conducted, with proportions ranging from 14.2% (in 2017) to 16.2% (in 2022). Training was provided on a monthly or quarterly basis by approximately 5% to 7% of the clinics, while no clinics reported offering weekly training in any year.

The proportion of participants in the cross-institutional incident reporting system increased from 44.5% (n = 1157) in 2017 to approximately 55% in 2019–2022 (Table 5).

The participation of hospitals in various cross-institutional incident reporting systems was examined over the period from 2017 to 2022 (Figure 1). There was a notable increase in participation in the hospital IRS, rising from 15.9% (n = 414) in 2017 to 23.5% (n = 567) in 2022 (*p* < 0.001). A significant increase was also observed in the rate of participation in CIRS Berlin, from 4.9% (n = 127) in 2017 to 6.7% (n = 161) in 2022 (*p* = 0.011) (Figure 2). The complete table can be viewed in Appendix A.

## 4. Discussion

### 4.1. Internal IRSs

The present study offers an overview of the implementation and use of internal and cross-institutional incident reporting systems (IRSs) in German hospitals between 2017 and 2022. The results indicate that participation in internal IRSs remained consistently high throughout the entire observation period, with a slight increase from 94% in 2017 to 96.7% in 2022. This consistency may suggest that German hospitals are largely compliant with the legal requirements for implementing IRSs. In Switzerland, 86% of the surveyed hospitals (84 of 98) used an internal IRS system on a voluntary basis as of 2010 [15]. Austria offers an insightful example through the findings of Sendlhofer et al. (2019), who investigated the implementation of the Critical Incident Reporting System (CIRS) in public hospitals within the federal state of Styria. Unlike mandatory systems, the CIRS in Styria operates on a voluntary basis, with the study demonstrating a progressive increase in reporting rates over a five-year period. However, the study also underscores persistent challenges, including underreporting, concerns regarding the potential compromise of anonymity, and the lack of standardised procedures across healthcare institutions. Despite these barriers, the study highlights the substantial potential of the CIRS to enhance organisational learning and patient safety, particularly when its implementation is bolstered by committed leadership and comprehensive, ongoing training initiatives [16]. This potential, however, contrasts with the broader adoption of incident reporting systems (IRSs) in Germany. Meanwhile, a 2016 survey in Austria found that only 64.1% of hospitals utilised an internal IRS [17]. The discrepancy in participation rates between these countries and Germany may be attributed to the fact that, unlike in Germany, an IRS is not legally required in Austria and Switzerland.

The proportion of hospitals that regularly conduct internal evaluations of their incident reports remained stable at approximately 85% throughout the entire period, which serves to reinforce the continuous use of these systems. Nevertheless, a number of hospitals still fail to conduct regular evaluations, which suggests that there are significant practical implementation challenges. One potential explanation for this phenomenon is that a considerable proportion of reports are incomplete and inaccurate, which has an adverse effect on their utility for medical incident research [18]. A noteworthy finding of this study is the relatively low frequency of staff training in the use of IRSs. Although on-demand training is a common practice in most hospitals, with an estimated 40% of hospitals offering this type of training, only a few hospitals provide regular training, typically on a monthly or quarterly basis. Evans et al. identified inadequate training as a significant barrier to the use of IRSs. The analysis revealed that employees who had received training on a reporting system were significantly more likely to be able to complete reports correctly and use the system effectively [19]. Moreover, it was demonstrated that the quality of training was a pivotal determinant of the frequency of system use [10,20]. It is therefore imperative that employees receive ongoing training to ensure that incident reporting systems are not only formally in place but also effectively integrated into clinical practice. The available evidence indicates that there is scope for improvement in this area, particularly with regard to the provision of regular training for staff, which is essential to cultivating a long-term institutional culture of vigilance and proactive management of incidents.

### 4.2. Cross-Institutional IRSs

The proportion of participants in cross-institutional incident reporting systems increased significantly from 2017 to 2019 (from 44.5% to 54.4%, *p* < 0.001), indicating a growing recognition of the advantages associated with such systems. Similarly, participation in hospital IRSs rose from 15.9% in 2017 to 23.5% in 2022, and the participation rate of the CIRS Berlin increased from 4.9% to 6.7%. These trends highlight an evolving engagement with IRSs in Germany, reflecting gradual improvements in adoption. However, despite these positive developments, the overall use of IRSs remains fragmented and decentralised, limiting their full potential for improving patient safety. In contrast, the UK has established a centralised incident reporting system, the National Reporting and Learning System (NRLS), resulting in higher participation rates and more effective incident analysis [21]. This centralised approach allows for consistent data collection and analysis, facilitating nationwide learning and targeted interventions. Switzerland has similarly achieved high participation rates in its voluntary IRS, supported by integration with hospital management and quality improvement initiatives [15]. Similarly, the Netherlands has implemented a national reporting system for anaesthesia, ensuring the nationwide documentation of critical events and the timely dissemination of information to all relevant professionals [22]. This facilitates learning from rare and hazardous events and improves the overall safety.

Despite these international successes, the findings of our study highlight that Germany’s decentralised approach, while offering some advantages in flexibility, lacks the necessary standardisation for comprehensive data analysis. Addressing this challenge by strengthening the connections between different systems through enhanced interoperability and technical improvements could provide a viable solution without requiring a complete shift to centralisation. Building on these considerations, a hybrid approach that retains decentralisation while leveraging technical improvements could offer a balanced solution. While centralised systems offer advantages in terms of comprehensive data analysis and enhanced patient safety, centralisation may not be the only solution for Germany. An alternative approach could involve further technical development and the improvement of interfaces to enable better data exchange and, at the very least, centralised analysis. This could help streamline the reporting process, reduce redundancy, and enhance the overall utility of incident reporting systems without requiring complete centralisation. Such advancements would facilitate interoperability and allow for more efficient evaluation of critical incidents across various platforms. Moreover, systematic surveillance can avert the recurrence of costly incidents when necessary [23].

### 4.3. Organisational and Structural Barriers to IRSs’ Effectiveness

The lack of significant improvement in regular IRS training over the years can be attributed to several factors. Resource constraints often pose a major barrier, as hospitals facing financial pressures tend to prioritise immediate operational needs over structured, ongoing training programs [24]. Consequently, training is frequently conducted on an ad hoc basis to address specific incidents rather than as part of a strategic, standardised approach. Additionally, the lack of prioritisation of IRS training within hospital management may reflect a broader focus on short-term goals such as cost reductions or efficiency improvements, sidelining the long-term benefits of fostering a robust safety culture [25].

Variations in hospital policies and organisational structures also contribute to inconsistencies in training practices across institutions. The geographic location (e.g., urban vs. rural hospitals) often influences the access to resources and the capacity to deliver regular training, with rural hospitals potentially facing greater challenges due to limited budgets and staffing shortages. Furthermore, competitive pressures in certain regions may lead hospitals to prioritise visible quality metrics over long-term investments like IRS training. While some hospitals have established comprehensive training programs, others lack the necessary processes or resources to sustain regular sessions [25].

The absence of a centralised authority to enforce standardised IRS training, as seen in countries with centralised systems like the UK, further exacerbates these challenges. For instance, the lack of standardisation in Germany’s decentralised system limits the potential for cohesive, nationwide improvements in reporting practices. This fragmentation became particularly evident during the COVID-19 pandemic, where the expected strain on healthcare systems resulted in a non-significant change in reporting rates compared to 2019. This suggests that the crisis further exposed limitations in existing quality management systems and reporting frameworks, which lacked the flexibility or prioritisation to adapt effectively. Addressing these barriers through financial incentives, standardised national guidelines, flexible e-learning platforms, and awareness campaigns could significantly enhance the regularity and effectiveness of IRS training programs. Moreover, suggested policies should focus on improving the interoperability between systems and supporting rural hospitals with targeted resource allocation. Future research could explore the relationship between staffing resources, geographic location, and hospital engagement with IRSs, providing deeper insights into how systemic barriers impact reporting practices and offering actionable strategies to enhance participation and training consistency.

### 4.4. Limitations

Our findings are based on the data reported by hospitals in structured quality reports, which may not fully capture the real-world usage of incident reporting systems in everyday clinical practice. Hospitals might formally report participation in IRSs, yet the actual depth of incident reporting and evaluation could be insufficient. Additionally, the absence of data for 2018 represents a further limitation, potentially affecting the completeness of the analysis over the studied period.

## 5. Conclusions

This study demonstrates that internal incident reporting systems (IRSs) are widespread in German hospitals, but their adoption has largely plateaued. Despite this, there is significant room for improvement, particularly in the areas of regular assessment and structured training of personnel. While participation in cross-institutional IRSs has increased since 2017, the system remains fragmented and decentralised. Rather than relying solely on centralisation, the development of robust technical interfaces and enhanced data exchange mechanisms could greatly improve the efficiency and interoperability of these systems. By enabling better data sharing and centralised analysis without requiring full system integration, Germany can harness the benefits of decentralised systems while still achieving comprehensive incident evaluation. Addressing gaps in training and evaluation practices is critical to unlocking the full potential of IRSs, ultimately advancing patient safety and elevating overall healthcare quality. Through these targeted improvements, IRSs can serve as a cornerstone for a safer and more reliable healthcare system.

## Figures and Tables

**Figure 1 healthcare-12-02386-f001:**
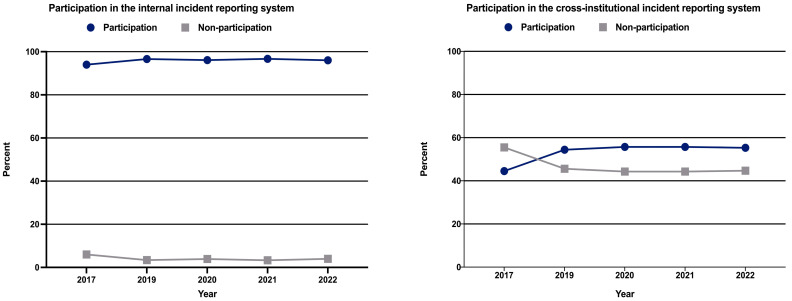
Participation in internal and cross-institutional incident reporting systems.

**Figure 2 healthcare-12-02386-f002:**
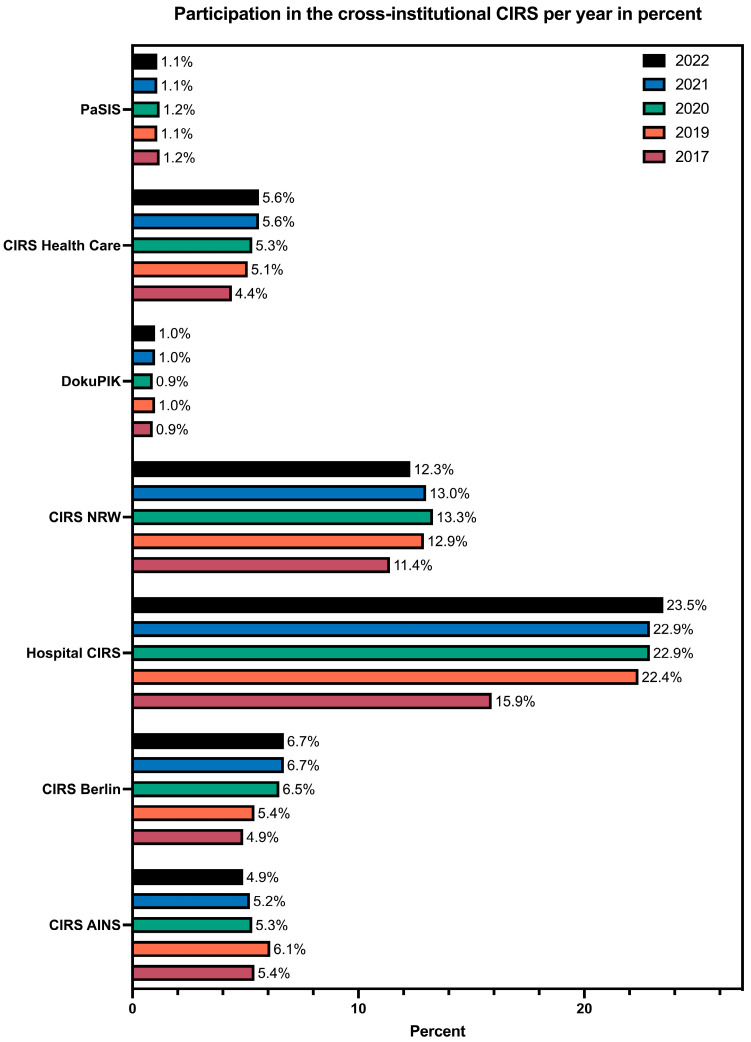
Participation in cross-institutional CIRSs from 2017 to 2022.

**Table 1 healthcare-12-02386-t001:** Regional distribution of participating hospitals across Germany.

Regions	2017 (%)	2019 (%)	2020 (%)	2021 (%)	2022 (%)
East Germany	186 (7.2%)	188 (7.2%)	193 (7.8%)	195 (7.9%)	190 (7.9%)
Northeast Germany	234 (9.0%)	256 (9.8%)	236 (9.5%)	243 (9.9%)	235 (9.8%)
Northwest Germany	277 (10.7%)	285 (10.9%)	282 (11.3%)	273 (11.1%)	259 (10.8%)
Middle Germany	299 (11.5)	290 (11.1%)	305 (12.2%)	302 (12.3%)	298 (12.4%)
West Germany	339 (13.0%)	336 (12.9%)	305 (12.2%)	301 (12.3%)	285 (11.8%)
West Germany	278 (10.7%)	278 (10.7%)	272 (10.9%)	268 (10.9%)	255 (10.6%)
Southwest Germany	215 (8.3%)	202 (7.8%)	187 (7.5%)	186 (7.6%)	185 (7.7%)
South Germany	235 (9.0%)	233 (9.0%)	230 (9.2%)	214 (8.7%)	229 (9.5%)
South Germany	285 (11%)	289 (11.1%)	267 (10.7%)	262 (10.7%)	262 (10.9%)
Southeast Germany	250 (9.6%)	246 (9.5%)	213 (8.6%)	211 (8.6&)	210 (8.7%)

**Table 2 healthcare-12-02386-t002:** Participation in the internal incident reporting system.

Year	Participation n (%)	Non-Participation n (%)	*p*-Value of Comparison to Following Year ^1^
2017	2442 (94.0%)	156 (6.0%)	<0.001
2019	2514 (96.6%)	89 (3.4%)	0.405
2020	2394 (96.1%)	96 (3.9%)	0.293
2021	2374 (96.7%)	81 (3.3%)	0.201
2022	2312 (96.0%)	96 (4.0%)	

^1^ Chi-square test (X^2^).

**Table 3 healthcare-12-02386-t003:** Evaluation of internal reports over the years.

Year	Implementation n (%)	Non-Implementation n (%)	*p*-Value of Comparison to Following Year ^1^
2017	2148 (82.7%)	450 (17.3%)	<0.001
2019	2243 (86.2%)	360 (13.8%)	0.259
2020	2118 (85.1%)	372 (14.9%)	0.466
2021	2107 (85.8%)	348 (14.2%)	0.721
2022	2058 (85.5%)	350 (14.5%)	

^1^ Chi-square test (X^2^).

**Table 4 healthcare-12-02386-t004:** Training of employees on the CIRS over the years.

Year	Training n (%)	No Training n (%)	*p*-Value of Comparison to Following Year ^1^
2017	1842 (70.9%)	756 (29.1%)	<0.001
2019	1964 (75.5%)	639 (24.5%)	0.214
2020	1841 (73.9%)	649 (26.1%)	0.580
2021	1832 (74.6%)	623 (25.4%)	0.742
2022	1787 (74.2%)	621 (25.8%)	

^1^ Chi-square test (X^2^).

**Table 5 healthcare-12-02386-t005:** Participation in the cross-institutional incident reporting system.

Year	Participation n (%)	Non-Participation n (%)	*p*-Value of Comparison to Following Year ^1^
2017	1157 (44.5%)	1441 (55.5%)	<0.001
2019	1416 (54.4%)	1187 (45.6%)	0.875
2020	1387 (55.7%)	1103 (44.3%)	0.989
2021	1368 (55.7%)	1087 (44.3%)	0.753
2022	1331 (55.3%)	1077 (44.7%)	

^1^ Chi-square test (X^2^).

## Data Availability

The data presented in this study are available on request from the corresponding author.

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
