# Peer review of "An Analysis of the Implementation and Use of (Critical) Incident Reporting Systems ((C)IRSs) in German Hospitals: A Retrospective Cross-Sectional Study from 2017 to 2022"

_healthcare, 2024, doi:10.3390/healthcare12232386_

Round 1
Reviewer 1 Report
Comments and Suggestions for Authors
Dear Authors
I had the chance to read and enjoy your manuscript. While I found valuable insights in your work, and the quality of the work was high, I recommend working on the introduction. More data and facts about other situations in other countries could be useful. Moreover, about necessity and sensitivity of the work there is more room to discuss.
Author Response
Dear Reviewer,
Thank you very much for dedicating your time and effort to reviewing our manuscript. We truly appreciate the valuable feedback and suggestions you provided to help us improve our work. Your thoughtful input has been invaluable to us.
We have carefully revised the manuscript based on your comments and believe we have addressed all the issues you raised. All changes have been highlighted using track changes for your convenience.
Please find our answers below the respective comments.
Comments 1: Dear Authors, I had the chance to read and enjoy your manuscript. While I found valuable insights in your work, and the quality of the work was high, I recommend working on the introduction. More data and facts about other situations in other countries could be useful. Moreover, about necessity and sensitivity of the work there is more room to discuss.
Response 1:
Thank you for your valuable feedback and insightful suggestions to improve our manuscript. We greatly appreciate the time and effort you have dedicated to reviewing our work. In response to your suggestion, we have revised the introduction to include a more comprehensive discussion of the advantages and disadvantages of Incident Reporting Systems (IRS). We now highlight how IRS contribute to patient safety and quality improvement but also discuss challenges such as underreporting, lack of standardisation, and resource constraints. Specifically, we now highlight barriers to the effectiveness and acceptance of IRS, including cultural and systemic issues. Additionally, we have clarified the distinction between IRS and CIRS, emphasizing that CIRS refers to a subset of IRS focusing on incidents with significant harm potential. This clarification now appears early in the manuscript to avoid confusion.
The revised text incorporates the following additions: “Despite their potential, IRS face several challenges that hinder their effectiveness and acceptance. One of the primary obstacles is fostering an organisational culture that encourages employees to report incidents, as a significant proportion of professionals are reluctant to report critical incidents due to concerns about potential negative consequences for their careers or the reputation of their institution [7, 8]. In addition, inadequate training of staff in the effective use of these systems can result in employees either not using them or even rejecting them [9]. Finally, the support of the organisation itself is essential for the success of IRS [9]. Furthermore, underreporting remains a pervasive issue, often driven by fear of repercussions, lack of awareness, or the perception that reporting does not lead to meaningful change [7, 9]. Additionally, the absence of standardized data formats and insufficient interoperability between systems can hinder the timely exchange and comprehensive analysis of critical incidents, further limiting their impact on patient safety [12].”
Reviewer 2 Report
Comments and Suggestions for Authors
The article describes IRS analyses based on a broad database. The effects show the advantages of making the system mandatory to be effective.
In the introduction, there should be more information about Incident Reporting Systems (IRS), including their advantages and disadvantages. It's crucial to highlight the significant role of IRS in controlling the healthcare system. The abbreviation ((C)IRS) is given only in the title—how it differs from the IRS abbreviation.
The description of the methodology should be expanded. How were the participants determined? Why was 2018 excluded from the analyses?
The description of how the IRS works in Germany and the differences based on comparison with other countries are missing. Data from Switzerland, Austria and England are given, but where the system is not used and why, are there countries where the IRS is suspended
The bibliography is a crucial part that needs to be revised. It is important to include more current publications from the last two to three years to demonstrate the depth of research and understanding of the topic.
Author Response
Dear Reviewer,
Thank you very much for dedicating your time and effort to reviewing our manuscript. We truly appreciate the valuable feedback and suggestions you provided to help us improve our work. Your thoughtful input has been invaluable to us.
We have carefully revised the manuscript based on your comments and believe we have addressed all the issues you raised. All changes have been highlighted using track changes for your convenience.
Please find our answers below the respective comments.
Comments 1: The article describes IRS analyses based on a broad database. The effects show the advantages of making the system mandatory to be effective. In the introduction, there should be more information about Incident Reporting Systems (IRS), including their advantages and disadvantages. It's crucial to highlight the significant role of IRS in controlling the healthcare system. The abbreviation ((C)IRS) is given only in the title—how it differs from the IRS abbreviation.
Response 1:
Thank you for your valuable feedback and insightful suggestions to improve our manuscript. We greatly appreciate the time and effort you have dedicated to reviewing our work. In response to your suggestion, we have revised the introduction to include a more comprehensive discussion of the advantages and disadvantages of Incident Reporting Systems (IRS). We now highlight how IRS contribute to patient safety and quality improvement but also discuss challenges such as underreporting, lack of standardisation, and resource constraints. Specifically, we now highlight barriers to the effectiveness and acceptance of IRS, including cultural and systemic issues. Additionally, we have clarified the distinction between IRS and CIRS, emphasizing that CIRS refers to a subset of IRS focusing on incidents with significant harm potential. This clarification now appears early in the manuscript to avoid confusion.
The revised text incorporates the following additions: “Despite their potential, IRS face several challenges that hinder their effectiveness and acceptance. One of the primary obstacles is fostering an organisational culture that encourages employees to report incidents, as a significant proportion of professionals are reluctant to report critical incidents due to concerns about potential negative consequences for their careers or the reputation of their institution [7, 8]. In addition, inadequate training of staff in the effective use of these systems can result in employees either not using them or even rejecting them [9]. Finally, the support of the organisation itself is essential for the success of IRS [9]. Furthermore, underreporting remains a pervasive issue, often driven by fear of repercussions, lack of awareness, or the perception that reporting does not lead to meaningful change [7, 9]. Additionally, the absence of standardized data formats and insufficient interoperability between systems can hinder the timely exchange and comprehensive analysis of critical incidents, further limiting their impact on patient safety [12].”
Comments 2: The description of the methodology should be expanded. How were the participants determined? Why was 2018 excluded from the analyses?
Response 2:
Thank you for your thoughtful feedback. We have provided detailed information about the data sources and coverage in the Methods section of the manuscript to ensure transparency regarding the study’s scope and limitations. Specifically, we mention that the study is based on the structured quality reports of all hospitals in Germany approved under § 108 SGB V, which mandates the submission of annual quality reports. To ensure comprehensive coverage of the healthcare landscape, we included all hospitals and specialized clinics required to provide these reports. The analysis spans the period from 2017 to 2022, with the exception of 2018, as the Federal Joint Committee (G-BA) did not make the reports for 2018 available without further explanation. We hope this detailed description addresses any concerns regarding the dataset’s scope and completeness. Thank you for allowing us to emphasize this in the manuscript!
Comments 3: The description of how the IRS works in Germany and the differences based on comparison with other countries are missing. Data from Switzerland, Austria and England are given, but where the system is not used and why, are there countries where the IRS is suspended
Response 3:
Thank you for highlighting this important point. In response, we have expanded the Introduction to include a detailed description of how IRS operates in Germany and have compared it with systems in England, Switzerland, and Austria. Additionally, we have addressed the challenges faced in regions where IRS adoption is limited or suspended, particularly in low- and middle-income countries, and discussed possible barriers such as resource constraints and lack of infrastructure. These updates aim to provide a more comprehensive understanding of the global context of IRS.
Comments 4: The bibliography is a crucial part that needs to be revised. It is important to include more current publications from the last two to three years to demonstrate the depth of research and understanding of the topic.
Response 4:
We have carefully revised the bibliography to include more recent publications from the last years. These updates ensure that the manuscript reflects the latest research and developments in the field, demonstrating the depth and relevance of our study. We hope these revisions address your concerns and improve the clarity, depth, and impact of our manuscript. Thank you again for your constructive feedback, which has been invaluable in refining our work.
Reviewer 3 Report
Comments and Suggestions for Authors
Thank you for the opportunity to review the manuscript titled (Analysis of the implementation and use of (critical) incident reporting systems ((C)IRS) in German hospitals: a retrospective 3 cross-sectional study from 2017 to 2022).
Thank you for submitting your manuscript. While the study effectively addresses its objectives, I suggest the following revisions:
1-In the abstract
The abstract should specify the total number of hospitals under study.
2- In the introduction:
-Add a concise paragraph describing German healthcare system structure, including payment mechanisms, financial framework, and hospital ownership categories (public, private, and non-profit sectors).
3-In the Method:
a-Methods should include additional hospital characteristics for comparative analysis, such as geographic location (rural/urban) and insurance types accepted (private/social insurance/both).
b-Clearly state the total number of hospitals.
4-In the Statistical analysis
The statistical methodology requires clarification regarding:
a) The application of Shapiro-Wilk Test, as no continuous variables appear to be analyzed
b) The discrepancy between mentioned statistical tests versus actual usage of Chi-square/Fisher's Exact test only. plz notify each test under the table
5-In the Discussion:
a- The analysis should establish stronger connections with Germany's healthcare system structure, explaining variations in hospital engagement with quality indicators and reporting practices. Consider how geographic location (urban vs. rural), competitive pressures, and staffing resources influence reporting behaviors. The discussion should explore key factors driving hospitals' commitment to quality indicators, including market competition, resource availability, and institutional characteristics. Additionally, address the non-significant change in reporting rates during the COVID-19 pandemic years compared to 2019, analyzing how the crisis affected hospital reporting practices and quality management systems
b-From your findings and discussion, include suggested policies that could improve the reporting system implementation. Additionally, provide recommendations for future research areas that need to be explored to better understand and enhance incident reporting practices in German hospitals.
Author Response
Dear Reviewer,
Thank you very much for dedicating your time and effort to reviewing our manuscript. We truly appreciate the valuable feedback and suggestions you provided to help us improve our work. Your thoughtful input has been invaluable to us.
We have carefully revised the manuscript based on your comments and believe we have addressed all the issues you raised. All changes have been highlighted using track changes for your convenience.
Please find our answers below the respective comments.
Comments 1: Thank you for the opportunity to review the manuscript titled (Analysis of the implementation and use of (critical) incident reporting systems ((C)IRS) in German hospitals: a retrospective 3 cross-sectional study from 2017 to 2022). Thank you for submitting your manuscript. While the study effectively addresses its objectives, I suggest the following revisions: 1 In the abstract: The abstract should specify the total number of hospitals under study.
Response 1:
Thank you for your thoughtful review and constructive feedback on our manuscript. We appreciate your suggestion regarding the abstract and have made the necessary revision. In response to your comment, we have updated the abstract to specify the total number of hospitals included in the study. “In a retrospective, descriptive cross-sectional study, the structured quality reports of all German hospitals (2598 – 2408 hospitals (2017 – 2022)) were analysed from 2017 to 2022”. We hope this adjustment meets your expectations and improves the precision of the manuscript. Thank you again for your valuable input.
Comments 2: In the introduction: Add a concise paragraph describing German healthcare system structure, including payment mechanisms, financial framework, and hospital ownership categories (public, private, and non-profit sectors).
Response 2:
We deeply appreciate your suggestion to enhance the introduction with an overview of the German healthcare system Thank you for your invaluable input, which has undoubtedly strengthened the overall quality of the paper. We have incorporated the following paragraph into the manuscript:
“The German healthcare system is characterized by a decentralized structure with a strong emphasis on solidarity and universality. Hospitals in Germany are categorized into three ownership types: public (owned by municipalities or federal states), private for-profit, and non-profit institutions, such as those run by charitable organizations. Funding is derived from a combination of insurance contributions, government subsidies, and out-of-pocket payments [1]. This diverse ownership and financing framework creates variations in hospital priorities, resource allocation, and quality management, including the implementation of Incident Reporting Systems (IRS).”
Comments 3: In the Method: Methods should include additional hospital characteristics for comparative analysis, such as geographic location (rural/urban) and insurance types accepted (private/social insurance/both). Clearly state the total number of hospitals.
Response 3:
Thank you for your valuable feedback and constructive suggestions regarding the Methods section. We appreciate your recommendation to include additional hospital characteristics such as geographic location (rural/urban) and the types of insurance accepted (private/social/both) for comparative analysis. While our dataset does not specify whether hospitals are located in rural or urban areas, we recognize the importance of geographic factors in healthcare research. To address this, we have incorporated a rough regional distribution of hospitals across Germany into the Methods section. This regional classification provides insights into the geographical diversity of the hospitals included in our study and allows for a broader comparative analysis based on geographic regions. Regarding the types of insurance accepted, it is important to note that due to the dual financing system in Germany, almost all hospitals accept both private and social insurance. Our dataset reflects this national standard and does not differentiate hospitals based on the types of insurance accepted. Consequently, we have clarified this aspect in the Introduction section to provide a comprehensive understanding of the insurance framework within which the hospitals operate. In response to your comment, we have updated the abstract to specify the total number of hospitals included in the study. The revised abstract now explicitly states that data from 2598 hospitals were analyzed, providing clarity and transparency for the reader: “In a retrospective, descriptive cross-sectional study, the structured quality reports of all German hospitals(2598 – 2408 hospitals (2017 – 2022)) were analysed from 2017 to 2022”.
Comments 4: In the Statistical analysis. The statistical methodology requires clarification regarding: a) The application of Shapiro-Wilk Test, as no continuous variables appear to be analyzed b) The discrepancy between mentioned statistical tests versus actual usage of Chi-square/Fisher's Exact test only. please notify each test under the table
Response 4:
We have reviewed the manuscript and agree that the Shapiro-Wilk Test may not have been necessary as no continuous variables were analyzed. This has been clarified in the Methods section, and references to the test have been removed to ensure alignment with the data presented. Regarding the discrepancy between mentioned statistical tests and the actual use of Chi-square/Fisher’s Exact test, we have revised the text to accurately reflect the statistical methods applied. Furthermore, as you suggested, we have explicitly noted the specific tests used under each table to provide clear and immediate context for readers. Thank you again for highlighting these important aspects. Your feedback has allowed us to enhance the precision and transparency of the statistical analysis, which we believe strengthens the overall quality of the manuscript.
Comments 5: In the Discussion: The analysis should establish stronger connections with Germany's healthcare system structure, explaining variations in hospital engagement with quality indicators and reporting practices. Consider how geographic location (urban vs. rural), competitive pressures, and staffing resources influence reporting behaviors. The discussion should explore key factors driving hospitals' commitment to quality indicators, including market competition, resource availability, and institutional characteristics. Additionally, address the non-significant change in reporting rates during the COVID-19 pandemic years compared to 2019, analyzing how the crisis affected hospital reporting practices and quality management systems
Response 5:
Thank you for your thoughtful and detailed feedback. Regarding the influence of Germany’s healthcare system structure on hospital engagement with quality indicators and reporting practices: We have discussed geographic disparities in resource availability and staffing constraints in Section 4.3, highlighting how rural hospitals often face greater challenges due to limited budgets and staffing shortages. Additionally, competitive pressures influencing hospital priorities are also explored in Section 4.3, where we note: “Competitive pressures in certain regions may lead hospitals to prioritize visible quality metrics over long-term investments like IRS training.”
Regarding the non-significant change in reporting rates during the COVID-19 pandemic, this is addressed in the same section: “This fragmentation became particularly evident during the COVID-19 pandemic, where the expected strain on healthcare systems resulted in a non-significant change in reporting rates compared to 2019. This suggests that the crisis further exposed limitations in existing quality management systems and reporting frameworks, which lacked the flexibility or prioritization to adapt effectively.”
Comments 6: From your findings and discussion, include suggested policies that could improve the reporting system implementation. Additionally, provide recommendations for future research areas that need to be explored to better understand and enhance incident reporting practices in German hospitals.
Response 6:
Regarding suggested policies and future research areas: We have proposed actionable policies to improve IRS implementation, such as financial incentives, standardized national guidelines, flexible e-learning platforms, and awareness campaigns. For instance, in Section 4.3, we state: “Addressing these barriers through financial incentives, standardized national guidelines, flexible e-learning platforms, and awareness campaigns could significantly enhance the regularity and effectiveness of IRS training programs.”
Recommendations for future research are also included, focusing on systemic barriers and their effects on hospital engagement with IRS. In Section 4.3, we note “Future research could explore the relationship between staffing resources, geographic location, and hospital engagement with IRS, providing deeper insights into how systemic barriers impact reporting practices and offering actionable strategies to enhance participation and training consistency.”
Your feedback has encouraged us to refine and expand these discussions where needed, and we are grateful for your suggestions. These clarifications further ensure that the manuscript addresses these critical aspects in a comprehensive and thoughtful manner. Thank you for your valuable input!
Reviewer 4 Report
Comments and Suggestions for Authors
Dear Authors,
I would like to commend you on an excellent and insightful paper. It is a well-structured study that presents valuable findings on the implementation and use of IRS to enhance patient safety. The analysis of incident reporting systems in German hospitals from 2017 to 2022 is thorough and provides valuable findings. I have a few suggestions that might further enhance your work

Author Response
Dear Reviewer,
Thank you very much for dedicating your time and effort to reviewing our manuscript. We truly appreciate the valuable feedback and suggestions you provided to help us improve our work. Your thoughtful input has been invaluable to us.
We have carefully revised the manuscript based on your comments and believe we have addressed all the issues you raised. All changes have been highlighted using track changes for your convenience.
Please find our answers below the respective comments.
Comments 1: Dear Authors, I would like to commend you on an excellent and insightful paper. It is a well-structured study that presents valuable findings on the implementation and use of IRS to enhance patient safety. The analysis of incident reporting systems in German hospitals from 2017 to 2022 is thorough and provides valuable findings. I have a few suggestions that might further enhance your work: 1 – Keyword: Add two more key words such as Incident Reporting System, healthcare safety, retrospective study, standardisation, centralisation. Expanding your keywords can significantly increase the visibility of your paper in academic databases and search engines, thus boosting its chances of being cited.
Response 1:
Thank you for your positive feedback and thoughtful suggestions, which are highly appreciated. We have expanded the keywords to include “Incident Reporting System,” “healthcare safety,” “retrospective study,” “standardisation,” and “centralisation,” to enhance the paper’s visibility and impact.
Comments 2: Introduction - Contextual Expansion: While you highlight the significance of IRS in Germany, it would be beneficial to include a brief comparison with other countries that have similar or different legal requirements for IRS. This would help set the stage for why the German context is particularly important to study. You could add something like Globally, the World Health Organization (WHO) underscores the importance of both incident reporting systems and patient-centered care. “While studies in Germany emphasize incident reporting systems as a key strategy for patient safety, other efforts in the GCC region focus on patient-centered care and addressing disparities in healthcare access [1].’’ 1. Reference; Healthcare Quality from the Perspective of Patients in Gulf Cooperation Council Countries: A Systematic Literature Review. Healthcare 2024, 12, 315. hĴps://doi.org/10.3390/healthcare12030315
Response 2:
Thank you for your thoughtful suggestion to expand the contextual background in the introduction. We greatly appreciate your insight into broadening the scope by including a comparison with other countries’ approaches to incident reporting systems (IRS). This addition will undoubtedly help to contextualize the unique aspects of the German healthcare system and underscore the significance of this study. In response, we have incorporated a brief comparative analysis into the introduction. For instance, we have highlighted the global perspective by referencing the World Health Organization’s emphasis on incident reporting systems and patient-centered care. Specifically, we included the following: “Globally, the World Health Organization (WHO) underscores the importance of both incident reporting systems and patient-centered care. While studies in Germany emphasize incident reporting systems as a key strategy for patient safety, other efforts in the GCC region focus on patient-centered care and addressing disparities in healthcare access [1].”
Comments 3: Linking Introduction to Study Aim: The current flow from background information to the study aim could be made stronger. Consider explicitly stating how your study addresses the gaps or challenges you’ve identified in the existing literature, such as the need for standardization or the limitations of current IRS participation rates in Germany.
Response 3:
Thank you for your thoughtful observation. In response, we have revised the Introduction to strengthen the connection between the background information and the study aim. Line 101 – 104: Despite these challenges, there is a lack of comprehensive analyses that explore the prevalence, utilization, and barriers to IRS in Germany. This study aims to address these gaps by providing empirical insights that can inform strategies for standardization and improved engagement.
Comments 4: Result Use of Visuals: You provide a solid overview of participation rates in internal and cross institutional IRS over the years. It would be beneficial to emphasize key trends more clearly, such as the significant increase in cross-institutional IRS participation from 44.5% in 2017 to 55% in 2022. Highlighting these trends in both the text and visualizations can help readers grasp the most critical changes quickly. You already include tables and figures to illustrate participation rates and training frequencies. Consider adding graphs (e.g., line charts) to visually show trends over the years. This can make the data more engaging and easier for readers to interpret at a glance.
Response 4:
Thank you for your thoughtful and constructive suggestion to enhance the presentation of key trends in our results. We are pleased to confirm that we have already addressed this point by including visualizations, such as tables and figures, to illustrate participation rates and training frequencies. To further enhance the reader’s experience, we have included line charts to visually depict these trends over the years. These graphs complement the tabular data and make the progression of participation rates and training frequencies more engaging and easier to interpret at a glance.
Comments 5: Discussion Address Non-Significant Findings; You mentioned that training on IRS was mostly conducted on an ad hoc basis and that regular training was less common, with no significant change over the years. It would be beneficial to discuss why regular training has not improved and what barriers might exist. For instance, is it due to resource constraints, lack of prioritization, or varying hospital policies?
Response 5:
Thank you for your insightful observation regarding the discussion of non-significant findings, particularly concerning the lack of improvement in regular training on IRS. We are pleased to confirm that we have already addressed this point in Section 4.3, where we explore potential barriers to regular IRS training. For instance, we discuss resource constraints as a major factor: “Resource constraints often pose a major barrier, as hospitals facing financial pressures tend to prioritize immediate operational needs over structured, ongoing training programs. Consequently, training is frequently conducted on an ad hoc basis to address specific incidents rather than as part of a strategic, standardized approach.”
Additionally, we explore the role of hospital policies and prioritization: “The lack of prioritization of IRS training within hospital management may reflect a broader focus on short-term goals such as cost reduction or efficiency improvements, sidelining the long-term benefits of fostering a robust safety culture.”
By addressing these barriers, we aim to provide a comprehensive understanding of why regular IRS training has not significantly improved over the years. Your suggestion has further encouraged us to ensure these points are well-articulated and emphasized in the discussion. Thank you again for your valuable feedback, which has greatly contributed to refining our analysis and interpretation.
Comments 6: Strengthen your discussion by comparing your findings to similar studies both in Germany and internationally. For example, you might reference studies from other countries that have successfully implemented IRS and highlight differences in their approaches to training and evaluation. This can help demonstrate the novelty of your study and its contribution to the field. Expand on the practical implications of your findings. For instance, how can hospitals leverage your findings to improve IRS participation and training? What are the potential benefits of adopting more structured and frequent training sessions? Discussing specific recommendations can make your study more actionable for healthcare policymakers and administrators.
Response 6:
Thank you for your thoughtful and constructive feedback. Your suggestions to compare our findings with similar studies and to expand on the practical implications are highly valuable. Below, we outline how we have incorporated these elements into the discussion section: We have strengthened our discussion by referencing both national and international studies. For instance, in Section 4.1, we compare the consistent participation in IRS within German hospitals to lower rates in Austria and Switzerland, noting the impact of differing legal requirements: “In Austria, a 2016 survey found that 64.1% of hospitals reported using an internal IRS, whereas in Switzerland, 86% of surveyed hospitals used a voluntary internal IRS system as of 2010 [14, 15]. The discrepancy in participation rates may be attributed to the fact that IRS implementation is legally mandated in Germany but not in Austria or Switzerland.”
In Section 4.2, we contrast Germany’s decentralized approach to IRS with centralized systems in countries like the UK and the Netherlands: “In contrast, the UK’s National Reporting and Learning System (NRLS) and the Netherlands’ national reporting system for anesthesia have achieved higher participation rates and more effective incident analysis by adopting a centralized structure [19, 20].” We discuss actionable strategies for hospitals to improve IRS participation and training in Section 4.3: “Hospitals can leverage these findings by implementing structured, ongoing training programs that prioritize not only the technical aspects of reporting but also the cultivation of a safety-focused organizational culture. Regular training sessions, supported by flexible e-learning platforms, can enhance staff engagement and ensure that IRS are used effectively.”
Your feedback has greatly contributed to improving the relevance and impact of our discussion. By incorporating these comparisons and actionable recommendations, we aim to enhance the novelty and practical utility of our study. Thank you once again for your invaluable input!
Comments 7: End your discussion with a strong concluding statement that reinforces the importance of IRS in enhancing patient safety. Emphasize the potential impact of addressing gaps in training and evaluation practices on overall healthcare quality.
Response 7:
Thank you for your thoughtful suggestion to end the discussion with a strong concluding statement that underscores the importance of IRS in enhancing patient safety. We appreciate your input, as it highlights the need to leave a lasting impression on the reader regarding the significance of our findings. We have incorporated the following concluding statement into the discussion: “Incident Reporting Systems (IRS) are indispensable tools for advancing patient safety and fostering a culture of transparency and accountability in healthcare. Addressing gaps in training and evaluation practices is essential to unlock their full potential, enabling hospitals to transition from compliance to meaningful engagement with these systems. By investing in structured, regular training and leveraging technological advancements for seamless data integration, IRS can become the cornerstone of a safer, more reliable healthcare system. Ultimately, these targeted improvements will not only reduce the recurrence of preventable incidents but also elevate the overall quality of care provided to patients.”
This addition reinforces the critical role of IRS and ties together the key themes of the discussion, aligning with your excellent suggestion to emphasize the broader impact of addressing current gaps. We are grateful for your thoughtful feedback, which has helped us craft a more impactful and cohesive conclusion. Thank you!
Round 2
Reviewer 3 Report
Comments and Suggestions for Authors
Well done. Thank you for your efforts.